# Surface-Available HER2 Levels Alone Are Not Indicative of Cell Response to HER2-Targeted Antibody–Drug Conjugate Therapies

**DOI:** 10.3390/pharmaceutics16060752

**Published:** 2024-06-02

**Authors:** Molly Major, Christine S. Nervig, Annette Gerland, Shawn C. Owen

**Affiliations:** 1Department of Molecular Pharmaceutics, College of Pharmacy, University of Utah, 30 South 2000 East, Salt Lake City, UT 84112, USA; 2Department of Medicinal Chemistry, College of Pharmacy, University of Utah, 30 South 2000 East, Salt Lake City, UT 84112, USA; 3Department of Biomedical Engineering, College of Engineering, University of Utah, 36 S. Wasatch Drive, SMBB 3100, Salt Lake City, UT 84112, USA

**Keywords:** antibody–drug conjugates, HER2 quantification, HER2 internalization, trastuzumab, antigen overexpression, trastuzumab–MMAE, trastuzumab–DM1

## Abstract

HER2-targeting therapies have advanced breast cancer treatment over the past decade. Clinically, eligibility for HER2 therapies is determined by assessing HER2 levels on tumor cell surfaces through immunohistochemistry or by gene regulation through fluorescence in situ hybridization. HER2 therapies are not always effective in patients with elevated levels of HER2, questioning whether the amount of HER2 is sufficiently predictive of patient outcomes. Additionally, the HER2-targeting antibody–drug conjugate (ADC) Enhertu^®^ was recently approved for metastasized HER2-low cancers, confirming the benefits of HER2 treatment for patients with low HER2 levels. To evaluate the correlation between HER2 levels and treatment efficacy, we quantified HER2 on eight cell lines using flow cytometry while simultaneously determining the toxicity of two HER2-targeting ADCs. Both HER2-high cell lines and HER2-low cell lines had significant toxicity responses to ADCs. We quantified HER2 internalization and found no correlation between HER2 levels and the percentage of internalization. We found a useful metric suggesting that a minimum number of HER2 receptors trafficked to lysosomes is sufficient to provide effective treatment. Our results indicate that the current standards of determining eligibility for HER2 therapy could limit patients’ access to effective treatment. In conclusion, HER2 levels are not wholly adequate to determine the response to ADC treatment.

## 1. Introduction

Human epidermal growth factor 2 (HER2) belongs to the family of transmembrane receptor tyrosine kinases, along with the epidermal growth factor receptor (EGFR), HER3, and HER4 [1,2]. This family of receptors is known to be involved in cell proliferation, survival, differentiation, and migration; the overexpression of these proteins in cancer is associated with tumor progression and poor prognosis [1,2]. Specifically, HER2 upregulation is considered an important indication of cancer aggressiveness; therefore, HER2 has become a target for cancer therapy. Currently, there are multiple forms of HER2-targeting therapies available in the clinic, including monoclonal antibodies (trastuzumab, pertuzumab, and margetuximab), tyrosine kinase inhibitors (lapatinib, neratinib, and tucatinib), and antibody–drug conjugates (ado-trastuzumab emtansine and trastuzumab deruxtecan) [3].

Antibody–drug conjugates (ADCs) are a class of biotherapeutics that combine the target specificity of monoclonal antibodies with the cytotoxic capabilities of small-molecule chemotherapeutics. These small-molecule cytotoxins are attached to the antibody scaffold through a linker that generally requires internalization and trafficking to the lysosome for intracellular drug release. While ADCs are in circulation, the drugs should remain attached to the antibody and have negligible cytotoxic capabilities [4,5]. ADCs are developed to target overexpressed antigens on cancer cell surfaces; the overexpression of these antigens, such as HER2, is thought to be the reason that ADCs target cancer cells selectively over healthy cells [6]. Currently, there are two FDA-approved HER2-targeting ADCs: ado-trastuzumab emtansine (Kadcyla^®^) and trastuzumab deruxtecan (Enhertu^®^). Both therapeutics are approved for HER2-positive metastatic breast cancer, defined as tumors with an immunohistochemical (IHC) score ≥ 3+ or an IHC score ≥ 2+ with evidence of gene amplification [7]. Enhertu^®^ was approved in 2022 for HER2-low breast cancers, which are defined by an IHC score = 1–2+ with no evidence of gene amplification; in April of 2024, Enhertu received tumor-agnostic approval for unresectable or metastatic HER2-positive solid tumors [8,9,10].

The approval of Enhertu^®^ for HER2-low breast cancer refutes the original idea that overexpression is essential for the selectivity of cancer cells. In addition, previous studies are contradictory on whether there is a direct correlation between ADC efficacy and target antigen expression. Barok et al. found no correlation between the HER2-binding capacity of cell lines with the efficacy of trastuzumab or the ADC trastuzumab–mertansine (T-DM1) on a panel of HER2-overexpressing cancer cells [11]. In their studies of ADCs targeting CD22, Li et al. found that the expression of CD22 was a poor predictor of ADC response in CD22-low cell lines [12]. Clinical studies by Baselga et al. evaluated T-DM1 treatment on patients with differing HER2 expression, resulting in only a 2.4-month longer progression-free survival (PFS) for patients with high HER2 expression (8.2-month PFS for HER2-low patients and 10.6-month PFS for HER2-high patients) [13]. Although patients with high HER2 levels had improved outcomes, it is striking that patients with low HER2 levels still had significant responses to the treatment. Nevertheless, studies by Shah and coworkers have demonstrated a strong correlation between HER2 expression and the efficacy of the HER2-targeting ADC trastuzumab–MMAE both in vitro and in vivo [14,15].

Based on reports of the varying efficacy of ADCs among HER2-positive cells, as well as their demonstrated efficacy on some HER2-low and HER2-negative cell lines, we investigated the correlation between ADC efficacy and the HER2 copy number on cell surfaces. We performed a quantitative analysis of HER2 occupancy on cell surfaces across eight different cell lines that represent a range of HER2 levels and assessed their viability upon treatment with HER2-targeting ADCs. Since intracellular internalization is critical for ADC drug release and activity, we also investigated the internalization and trafficking levels of HER2–antibody complexes to lysosomes after treatment with trastuzumab to understand the impact of HER2 trafficking on therapeutic efficacy.

## 2. Materials and Methods

### 2.1. Reagents

Cell lines were obtained from ATCC, and reagents were purchased from ThermoFisher Scientific^®^ (Waltham, MA, USA). BT-474 cells were cultured in Roswell Park Memorial Institute (RPMI) 1640 medium (ATCC modification, Gibco™ (Waltham, MA, USA)) supplemented with 20% FBS (Gibco™) and 1× Antibiotic-Antimycotic Solution (Caisson Labs (Smithfield, UT, USA), #ABL02). MCF-7 cells were cultured in Minimum Essential Medium (MEM) (Earle’s salts, L-glutamine, Caisson Labs) supplemented with 10% FBS and 1% Antibiotic-Antimycotic Solution. MDA-MB-231 and MDA-MB-453 were cultured in Dulbecco’s Modified Eagle Medium (DMEM) (high glucose, pyruvate, Gibco™) supplemented with 10% FBS and 1× Antibiotic-Antimycotic Solution. NCI-N87 and OVCAR-3 cells were cultured in Roswell Park Memorial Institute (RPMI) 1640 medium (ATCC modification, Gibco™) supplemented with 10% FBS (Gibco™) and 1× Antibiotic-Antimycotic Solution. SK-BR-3 and SKOV-3 were cultured in McCoy’s 5A (Modified) Medium (Gibco™) supplemented with 10% FBS and 1× Antibiotic-Antimycotic Solution. All cell lines were incubated at 37 °C and 5% CO_2_ and used at passage 12 or below, except for the prolonged, passage-number-dependent studies on SK-OV-3 and NCI-N87 cell lines. Herceptin^®^ (trastuzumab) was purchased from Genentech, Inc. (South San Francisco, CA, USA) SMCC-DM1 and vcMMAE linker–drug constructs were purchased from Toronto Research Chemicals (Toronto, ON, Canada) and MedChemExpress (Monmouth Junction, NJ, USA), respectively, and used as received. Alexa Fluor™ 647 NHS Ester was purchased from Invitrogen (Waltham, MA, USA) and used as received. Quantum™ Alexa Fluor^®^ 647 MESF calibration beads were purchased from Bangs Laboratories (Fishers, IN, USA), and all flow studies were performed on the same batch of beads to avoid variability between lots. Mass spectrometry was performed on an ACQUITY UPLC I-Class with a Xevo G2-S QToF mass spectrometer (Waters Corporation (Milford, MA, USA)). Flow cytometry was performed on a BD FACSCanto™ II Flow Cytometry System and analyzed using FlowJo™ v10.8.1.

### 2.2. Graphing and Statistical Analysis

All statistical analyses were performed using GraphPad Prism version 10.2.1 for Windows (GraphPad Software, San Diego, CA, USA, www.graphpad.com). Differences among groups were assessed by ordinary one-way with Tukey’s multiple comparisons post hoc test to identify statistical differences among three or more treatments. Alpha levels were set at 0.05, and a *p*-value of ≤0.05 was set as the criterion for statistical significance. Graphs are annotated, where *p*-values are represented as ** *p* < 0.01, *** *p* < 0.001, and **** *p* < 0.0001. Absolute IC_50_ curves were fit in GraphPad Prism using non-linear regression from normalized data and are reported as means with 95% confidence intervals. Statistical significance was determined by first converting the log of the 95% CI to the standard deviation (SD) with the following equation and analyzing the results using one-way ANOVA with Tukey’s multiple-comparisons post hoc test.
$$SD=\sqrt{N}*\left(\frac{\log upper\;limit-\log lower\;limit}{t\mathrm{-value}*2}\right)\;$$
where the *t*-value is calculated as t-value=TINV(1−0.95,N−1).

The harmonic mean was calculated in GraphPad Prism version 10.2.1 to better represent the non-Gaussian distribution of HER2 expression levels on SKOV3 cell surfaces.

### 2.3. Synthesis of T-AF647 for HER2 Labeling

Alexa Fluor™ 647 was conjugated to trastuzumab using the following procedure. Trastuzumab (5.0 mg, 34 nmol, 1 equiv) was buffer-exchanged into amine conjugation buffer (10 mM sodium bicarbonate, pH 8.5) using a 10 kDa MWCO Amicon^®^ Ultra 0.5 mL centrifugal filter (EMD Millipore (Burlington, MA, USA)) and adjusted to 2.5 mg/mL. DMSO (200 µL, 10% *v*/*v*) was added slowly, followed by Alexa Fluor™ 647 NHS Ester (85 µL, 136 nmol, 2 mg/mL in DMSO, 4 equiv), and the reaction mixture was stirred gently in the dark for 60 min at 21 °C. The antibody–fluorophore conjugate T-AF647 was purified over a Zeba™ Spin Desalting Column (ThermoFisher Scientific, #89890) and dialyzed against DPBS with 1 mM EDTA at pH 7.4 using 10 kDa MWCO Slide-A-Lyzer™ Dialysis Cassettes (ThermoFisher Scientific) and characterized by mass spectrometry as described. According to the literature protocol [16], the degree of labeling (DOL) was also estimated by UV-vis using the following equation using extinction coefficients of trastuzumab ($\varepsilon_{280nm}$ = 225,000 M^−1^ cm^−1^) and AF647 ($\varepsilon_{650nm}$ = 239,000 M^−1^ cm^−1^) with a correction of 3% for the absorbance of AF647 at 280 nm.
$$DOL=\frac{\left(\frac{{Abs}_{650nm}}{\varepsilon_{650nm}}\right)}{\left(\frac{{Abs}_{280nm}\;-\;0.03\;\times \;{Abs}_{650nm}}{\varepsilon_{280nm}}\right)}$$

### 2.4. Synthesis of T-DM1 through Lysine Conjugation

According to the literature protocol [16], SMCC-DM1 (Toronto Research Chemicals) was conjugated to trastuzumab using the following procedure. Trastuzumab (5.25 mg, 36 nmol, 1 equiv) was adjusted to 3.0 mg/mL in borate buffer (50 mM Boric Acid, 50 mM NaCl, 2 mM EDTA, pH 8.0). While stirring, DMA (10% *v*/*v*) was added, followed by the slow addition of SMCC-DM1 (16.1 µL, 320 nmol, 20 mM in DMA, 9 equiv). The reaction mixture was stirred for 60 min at 21 °C and was purified over a Zeba™ Spin Desalting Column (ThermoFisher Scientific, #89890). The resulting ADC was dialyzed against DPBS (Gibco™) with 1 mM EDTA at pH 7.4 using 10 kDa MWCO Slide-A-Lyzer™ Dialysis Cassettes (ThermoFisher Scientific) and characterized by mass spectrometry as described.

### 2.5. Synthesis of T-MMAE through Native Cysteine Residues

Monomethyl auristatin E (MMAE) was conjugated to native cysteine residues using the following procedure. Trastuzumab (2 mg, 13.6 nmol mAb, 1 equiv) was adjusted to 2.0 mg/mL in DPBS with 1 mM EDTA at pH 7.4 (1 mL). Bond-Breaker™ TCEP Solution (10.9 µL of 12.5 mM, 136 nmol, 10 equiv, ThermoFisher Scientific^®^, #77720) was added to a gently stirring ADC solution to reduce disulfides. The reaction mixture was stirred at 37 °C for 1 h. The reaction mixture was cooled to 21 °C before the slow addition of DMSO (99 µL, 10% *v*/*v*). Next, vcMMAE (13.6 µL of 10 mM, 136 nmol, 10 equiv) was added slowly while watching for precipitation or cloudiness. The conjugation reaction proceeded at 21 °C for 1 h and was purified over a Zeba™ Spin Desalting Column (ThermoFisher Scientific, #89890). The resulting ADCs were dialyzed against DPBS with 1 mM EDTA at pH 7.4 using 10 kDa MWCO Slide-A-Lyzer™ Dialysis Cassettes (ThermoFisher Scientific) and characterized by mass spectrometry as described.

### 2.6. Mass Spectrometry (MS) Analysis of Antibody Conjugates

Prior to MS analysis, 50 µg of antibody or ADC at 1 mg/mL was deglycosylated by combining 5 µL of glycol buffer 1 (New England BioLabs (Ipswich, MA, USA)) and 5 µL of Remove-iT EndoS (1:10 dilution in PBS, 20,000 units/mL, New England BioLabs) and incubating at 37 °C for 1 h. Following deglycosylation, the sample was buffer-exchanged into ammonium acetate buffer (50 mM ammonium acetate, pH 7.0) using a 10 kDa Amicon^®^ Ultra 0.5 mL centrifugal filter (EMD Millipore). Intact LC/MS analysis was performed on an ACQUITY UPLC I-Class with a Xevo G2-S QToF mass spectrometer (Waters Corporation.

For lysine conjugations: Samples were further desalted with a MassPREP micro desalting column (Waters Corporation) using a 6 min linear gradient run at a flow rate of 0.3 mL/min, 80 °C. The gradient was programmed as follows: 5% B from 0 to 2 min, 5–90% B from 2 to 5 min, and then 90–5% B from 5 to 6 min. Mobile phase A was 0.1% (*v*/*v*) formic acid in HPLC-grade H_2_O, and mobile phase B was 0.1% (*v*/*v*) formic acid in acetonitrile.

For cysteine conjugations that are unstable in denaturing conditions, samples were further purified on a Waters ACQUITY UPLC Protein BEH SEC Column (200 Å, 1.7 µm, 4.6 mm × 150 mm) using a 10-min isocratic method with a flow rate of 0.3 mL/min and a mobile phase of 100 mM ammonium acetate.

The mass spectrometer was operated in positive electrospray ionization (ESI) mode. The capillary voltage was 3 kV, and the sampling cone voltage was 150 V. The source temperature was 150 °C, and the desolvation temperature was 500 °C. The source desolvation gas flow and cone gas flow were 800 L/h and 10 L/h, respectively. The recorded mass spectra were combined and deconvoluted using vendor-supplied MassLynx 4.1 (Waters Corporation), and peak intensities from deconvoluted spectra were used to derive the relative proportion of species in each sample. The drug-to-antibody ratio (DAR) for each species peak was identified by dividing the difference between the MW of the peak and the unmodified trastuzumab peak (~145 kDa) by the observed MW of the payload/linker species. The following analysis of MS results assumes equal ionization efficiencies between various DAR species of the ADC. The relative peak intensity of each DAR species was calculated by dividing the peak area of that species by the total peak area of that sample. The average DAR for each ADC was calculated by summing the product of each DAR species multiplied by its relative peak intensity within a sample (weighted average). All mass spectra are shown in Appendix A.

### 2.7. Flow Cytometry Assays for HER2 Quantification

Cells were split into three T-75 cm^2^ Vented Tissue Culture Flasks (Celltreat^®^ (Pepperell, MA, USA)) in indicated cell culture media supplemented with FBS and Antibiotic-Antimycotic Solution and allowed to grow to confluence. From each flask, cells were detached by incubating with 5 mL of Trypsin-EDTA (0.25%, Gibco™) for 5 min at 37 °C, pelleted by centrifugation, and resuspended in cell culture media. In triplicate, 1 × 10^6^ cells were fixed with 500 µL of 4% paraformaldehyde for 5 min at 21 °C and washed two times with 500 µL of DPBS. Cell pellets were resuspended in 500 µL of 100 nM T-AF647 in DPBS and incubated for 60 min at 0 °C, mixing by inversion every 15 min. Cells were washed three times with 500 µL of DPBS. Cells were resuspended in 500 µL of DPBS for flow cytometry performed on a BD FACSCanto™ II Flow Cytometry System and analyzed using FlowJo™ v10.8. Flow data were gated first with FSC-A by SSC-A and then gated for single cells with FSC-W by FSC-A. AF647 fluorescence intensity for all gated single cells was exported for further analysis in GraphPad Prism.

AF647 fluorescence intensity was calibrated using the Quantum™ Alexa Fluor^®^ 647 MESF kit from Bangs Laboratories according to the manufacturer’s recommended protocols. The geometric mean of AF647 intensity was analyzed by linear regression to establish a calibration curve for comparison with the T-AF647-labeled cells. Bead calibration was performed alongside all cell experiments to account for day-to-day variability. The MESF value of each cell was divided by the FAR of T-AF647 to determine the number of antibodies bound to HER2 on the cell surface, which directly correlates to the number of HER2 receptors on the cell surface, with the assumption that only one trastuzumab is bound per HER2 receptor [17]. Receptor saturation levels on cells were determined using a gradient of T-AF647.

### 2.8. In Vitro Cytotoxicity Assays

Cells were split into three T-75 cm^2^ Vented Tissue Culture Flasks (Celltreat^®^) in indicated cell culture media supplemented with FBS and Antibiotic-Antimycotic Solution and allowed to grow to confluence. From each flask, cells were detached by incubating with 5 mL of Trypsin-EDTA (0.25%, Gibco™) for 5 min at 37 °C, pelleted by centrifugation, and resuspended in cell culture media. Cells were seeded in triplicate at 2000 cells/well in tissue culture-treated 96-well plates (Celltreat^®^) and allowed to adhere for 24 h at 37 °C, 5% CO_2_. After adherence, media were removed and replaced with ADCs at varying concentrations in cell culture media in triplicate. Cells were incubated with the treatment for 5 days at 37 °C, 5% CO_2_. Cell viability was determined by removing the treatment solution and replacing it with 1:6 CellTiter 96^®^ AQ_ueous_ One Solution (Promega (Madison, WI, USA), G3580) in phenol-red-free RPMI-1640 media with 10% FBS and 1% Antibiotic-Antimycotic solution. According to the manufacturer’s protocols, the assay was allowed to develop at 37 °C, 5% CO_2_ for 3–4 h, and absorbance at 490 nm was recorded using a Cytation™ 3 plate reader (BioTek Instruments (Winooski, VT, USA)). IC_50_ values of cytotoxicity for ADCs were determined for normalized data using logistic non-linear regression analysis with GraphPad Prism software. Curve fits were constrained as needed to a lower bound of greater than 0% viability and an upper bound of less than 100% viability.

### 2.9. Confocal Microscopy Imaging

Cells were seeded in chamber slides at a concentration of 25,000 cells/mL and left to adhere overnight. The following day, the cells were treated with 100 nM of trastuzumab (Genentech, NDC 50242-132-01, Lot # 3581428) and incubated for 1, 24, or 48 h. Cells were then washed with PBST and fixed with 4% paraformaldehyde (PFA); for the 1 h treatments, the cells were fixed with PFA prior to trastuzumab treatment. Next, the wells were blocked with 1% bovine serum albumin in PBST. Primary staining was carried out with a 1:300 dilution of mouse anti-LAMP-2 (Santa Cruz Biotechnology (Dallas, TX, USA), sc-18822, Lot # A2319) and 100 nM of trastuzumab, which were left to sit at room temperature for 1 h. After washing, secondary stains were completed with a 1:250 dilution of goat anti-mouse antibody labeled with AlexaFluor-488 (Invitrogen, A32723, lot # TF266577) and a 1:400 dilution of goat anti-human antibody labeled with AlexaFluor-647 (Life Technologies (Carlsbad, CA, USA), A21445 Lot # 1495823). After 1 h, cells were washed and treated with a mounting medium containing 4′,6-diamidino-2-phenylindole (DAPI), covered with coverslips, and then allowed to sit at room temperature for 15 min. Imaging was performed on a LEICA SP8 Confocal Microscope equipped with LASX software version 3.5.7.23225 at the Cell Imaging Core at the University of Utah. For each cell line and treatment time, 2 replicates were made, and 3 random areas were imaged with at least 25 Z-stacks per image. Image processing was performed with IMARIS version 10.0.1 software. The number of HER2 receptors internalized is based on the percentage of total HER2 determined by flow cytometry.

## 3. Results and Discussion

### 3.1. Quantification of HER2 Expression on Representative Cancer Cell Lines

We quantified HER2 surface levels by constructing a fluorophore-labeled antibody and measuring the fluorescence intensity on fixed cells through flow cytometry. To construct the HER2-targeting fluorophore probe, we conjugated Alexa Fluor™ 647 (Invitrogen™) to trastuzumab. The fluorophore–antibody ratio (FAR) of the trastuzumab–Alexa Fluor™ 647 construct (T-AF647) was determined to be 3.6 through UV-Vis and verified by mass spectrometry (Appendix A) [18].

Using T-AF647 and calibrating fluorescence intensity with fluorescent polystyrene beads (Bangs Laboratories, Appendix A) as standards, we quantified cell-surface HER2 levels by flow cytometry (Figure 1, Table 1). The results show that cell lines traditionally termed “HER2-positive”, “HER2-low”, and “HER2-negative/null” encompass a range of average HER2 expression levels. In the HER2-positive cell lines (BT-474, SK-BR-3, NCI-N87, and SK-OV-3), the average HER2 expression level varied from 1.42 × 10^6^ HER2 receptors per cell to almost double that at 2.03 × 10^6^ HER2 receptors per cell. Within the HER2-low (MDA-MB-453 and OVCAR3) and HER2-negative cell lines (MCF-7 and MDA-MB-231), we observed a range of 2.81 × 10^4^ to 27.5 × 10^4^ HER2 receptors per cell. In addition, each cell line follows a different non-Gaussian distribution pattern.

Although cell lines are binned into categories of HER2-positive and HER2-negative, the HER2 expression levels between cell lines within their respective categories are significantly different. In fact, we found that most cell lines were statistically different from every other cell line with *p*-values less than 0.0001, except for MCF-7s and MDA-MB-231, which were not found to have any statistical difference, and SK-BR-3 and NCI-N87, which were found to have a *p*-value of 0.0014.

### 3.2. Response of Cancer Cell Lines to Trastuzumab Antibody–Drug Conjugates

In parallel to the quantification of HER2 receptors, we performed viability assays on cells harvested on the same day to determine the response to two different HER2-targeting ADCs with either monomethyl auristatin E (MMAE), a tubulin polymerization inhibitor [5], or emtansine (DM1), a maytansine derivative and a second microtubule inhibitor [19], to produce trastuzumab–monomethyl auristatin E (T-MMAE) and trastuzumab–emtansine (T-DM1), respectively. In ADC synthesis, chemical linkers are added to small-molecule drugs to facilitate conjugation to the antibody and to provide a mechanism for drug release after cellular internalization. MMAE was modified with a protease-cleavable valine–citrulline linker and conjugated through Michael addition of a terminal maleimide moiety to native cysteine thiol groups in trastuzumab to give the desired T-MMAE construct. DM1 was conjugated through a non-cleavable, heterobifunctional cross-linker (SMCC) to native lysine residues of trastuzumab to yield the T-DM1 construct [5]. For the ADC toxicity assays, cells were treated with T-MMAE or T-DM1 at a range of concentrations. Cell viability was measured through an MTS assay following 5 days of treatment (Appendix A).

Within their categories (HER2-positive, HER2-low, and HER2-negative), we observed a variety of treatment responses. Specifically, SK-OV-3 and BT-474 cells are deemed HER2-positive and contain the same order of magnitude of HER2 receptors per cell; however, we found that SK-OV-3 cells are less sensitive to T-MMAE treatment than BT-474 cells, with a 35× lower IC_50_. Interestingly, HER2-targeting therapies show efficacy against HER2-low cells. MDA-MB-453 cells (HER2-low) were more sensitive to both HER2-targeting therapies than the HER2-positive cell lines SK-OV-3 and NCI-N87 despite having 5 and 7 times lower (respectively) HER2 expression levels on the cell surface. It is also evident that the drug and/or linker conjugated to trastuzumab is relevant in determining treatment response, as seen in OVCAR3, SK-OV-3, BT-474, and SK-BR-3 cells, which had an order of magnitude difference between their T-DM1 and T-MMAE IC_50_ values. All HER2-positive cells had IC_50_ values with the same magnitude (10^−11^ M). HER2-negative and HER2-low cell lines varied from 33,000 (MDA-MB-231) to 306,000 (MDA-MB-453) HER2 receptors per cell, though this was not indicative of their response to treatment, with IC_50_ values varying from 10^−9^ to 10^−11^ M (Table 1).

While conducting our HER2 quantification experiments, we observed a non-Gaussian distribution response for the SK-OV-3 cell line and, therefore, sought to investigate further. At later passage numbers, we discovered an oscillating distribution pattern within the HER2 expression levels and corresponding oscillating IC_50_ values for both treatments, as shown in Figure 2. The harmonic mean of these passage numbers numerically demonstrates the visual pattern observed in the violin plots and the toxicity of the HER2-targeted treatments. Next, we investigated whether this pattern was also relevant in another HER2-positive cell line, NCI-N87, and we discovered that while there was a slight increase in HER2 expression levels with increasing passage number, NCI-N87s followed the same distribution pattern and did not exhibit significant differences in treatment response. We suspect that this oscillating pattern is unique to a cell-cycle-related process in the SK-OV-3 cell line. However, due to these results, it is reasonable to suggest that the determination of cell-surface protein levels and the toxicity of treatments be conducted under the same passage number and ideally from the same culture flask if the direct correlation between the surface receptor count and treatment is desired.

HER2 expression levels are typically determined clinically at the protein level with immunohistochemistry or at the DNA level with fluorescence in situ hybridization [7,10,20]. Higher HER2 levels qualify patients for HER2-targeted treatment, including ADC therapy; however, the approval of Enhertu^®^ for HER2-low breast cancer opens a new group of patients to potential therapy who lack overexpression of the target antigen. Due to this, we sought to determine whether HER2 levels alone are enough to determine treatment effectiveness for both HER2-positive and HER2-low cancers. A scatterplot of IC_50_ values versus the number of HER2 receptors shows no linear or exponential correlation for either ADC, indicating that the cell-surface HER2 level alone is not determinative of the response to treatment (Figure 3).

### 3.3. Quantification of Internalization of Trastuzumab ADCs

As the cell-surface HER2 quantity alone is not determinative of the response to treatment, we sought to determine whether the level of internalization may be more predictive. To investigate the variance in responses to HER2-targeted treatments, we conducted internalization experiments on all of our cell lines to investigate whether the amount of HER2 trafficked to the lysosomes of cells played an important role in IC_50_ values. We quantified internalized HER2 levels at three different time points, i.e., 1 h, 24 h, and 48 h, due to the postulated effect that trastuzumab induces the internalization of HER2 [14,21]. After measuring the percentage of HER2 colocalized with the lysosomes of the cells, we were able to determine a direct count based on the numerical value of HER2 established for each cell in the previous experiments (Figure 4). A comparison of the number of HER2 receptors internalized into the lysosomes to the efficacy of the ADCs revealed an apparent threshold for the efficacy of HER2-targeted therapies.

After roughly 25,000 HER2 proteins are internalized into the lysosomes, as observed in the HER2-low cell line MDA-MB-453, along with all HER2-positive cell lines, the efficacy of both HER2-targeting ADCs remains very similar in the sub-nanomolar IC_50_ range (see Figure 4 and Table 1). This effect seems to be independent of the drug mechanism. However, it is noted that the HER2-positive cell line SK-OV-3 has more drug dependence than other HER2-positive cell lines, which may be due to the variable HER2 expression levels observed in the HER2 quantification process (Figure 2). The HER2-low cell line MDA-MB-453 has lower IC_50_ values and HER2 levels than the HER2-positive cell lines but a higher percentage of internalized HER2 after 48 h (21%), resulting in more internalized HER2 proteins compared to SK-OV-3, BT-474, and NCI-N87 cell lines (Figure 4). This phenomenon logically explains the higher efficacy of HER2-targeted therapies on MDA-MB-453 cells over SK-OV-3, BT-474, and NCI-N87 despite having over 1,000,000 fewer HER2 receptors. This same pattern is observed in the HER2-negative and HER2-low cell lines. MDA-MB-231 has the lowest HER2 counts along with the lowest IC_50_ values for both HER-targeting treatments when compared to MCF-7 and OVCAR3 cell lines. At 24 h of trastuzumab treatment, MD-MB-231 cells have a much higher percentage of HER2 internalized and a higher number of HER2 receptors colocalized with the lysosomes, which is likely the reason for the increased efficacy. It is also likely that under the 25,000-internalized-HER2 threshold, the drug mechanism and/or linker composition plays an important role, as evidenced by the varying IC_50_ values between the HER2-targeting treatments in MCF-7 and OVCAR3 cell lines.

Considering the findings of the internalization studies, we investigated whether there was a correlation between HER2 internalized into the lysosome and the efficacy of the HER2-targeting ADCs. When plotted on a log–log scale, there is a general trend with an increasing number of HER2 receptors internalized resulting in lower IC_50_ values (Figure 5). This study suggests that the efficacy of HER2-targeting ADCs relies on both the number of HER2 proteins expressed and the level of HER2 internalization. Importantly, if HER2 expression is at a high level, defined here as over 1,000,000 per cell, low percentages of internalization can be overcome by the sheer number of receptors. However, with HER2-low and HER2-negative cell lines, higher internalization levels can compensate for low HER2 surface levels and result in an effective response, as seen with the MDA-MD-453 cell line.

## 4. Discussion

This study reveals a logical explanation for the apparent efficacy of HER2-targeted treatments in HER2-low cell lines. The combination of HER2 levels on the cell surface and internalization rates determine the effectiveness of the treatment. We observed that internalization rates are not a factor when it comes to HER2-positive cell lines, as their overabundance of HER2 (in the millions of receptors per cell) overcomes the need for efficient lysosomal trafficking of ADC-bound HER2 complexes. However, upon evaluating the HER2-low cell lines, MDA-MB-453 and OVCAR3, we found that internalization and trafficking rates played a major factor in the toxicity of the two ADCs. Overall, the number of HER2 receptors that are trafficked to the lysosomes for degradation determines the efficacy of an ADC, which can be described as the ratio of internalization rate to the number of HER2 receptors on the cell surface. Thus, we conclude that internalization rates are important for evaluating ADC targets and could compensate for lower expression rates compared to those of HER2-positive cancers.

In this work, we developed an assay using flow cytometry to analyze the exact number of HER2 receptors on the cell surface. The straightforward preparation for this assay allows for a detailed analysis of HER2 receptor expression and the distribution of expression within a cell line. Through this assay, the genetic drift of HER2 expression within a cell line can be easily quantified, as observed with the SK-OV-3 and NCI-N87 cell lines. We analyzed over 130,000 cells per cell line, giving us the utmost confidence in our results and the determination of average HER2 expression per cell of each cell line.

Elucidating a quantitative correlation between the cancer phenotype and therapeutic efficacy is clinically advantageous. Traditionally, for HER2 cancer, treatment decisions have been based on the overexpression of HER2. These guidelines are corroborated by detailed studies from Shah and co-investigators, which show a strong relationship between high-HER2-expressing cells and efficacy [14,15]. Here, we expand the literature to better understand the correlation between the antigen level and therapeutic efficacy in HER2-low and HER2-negative cell lines. Our results suggest that HER2 levels alone do not encompass the potential efficacy of HER2-targeted ADC treatment across the spectrum of HER2 expression levels. Results using a range of cells and using ADCs with two different drugs highlight the importance of antibody trafficking and antibody processing, as well as the pharmacodynamic sensitivity to ADC payloads. A more comprehensive patient assessment may be important in predicting the response to HER2-targeting ADCs, and developing a simple assay to determine internalization from patient samples is essential for personalizing patient care.

## Figures and Tables

**Figure 1 pharmaceutics-16-00752-f001:**
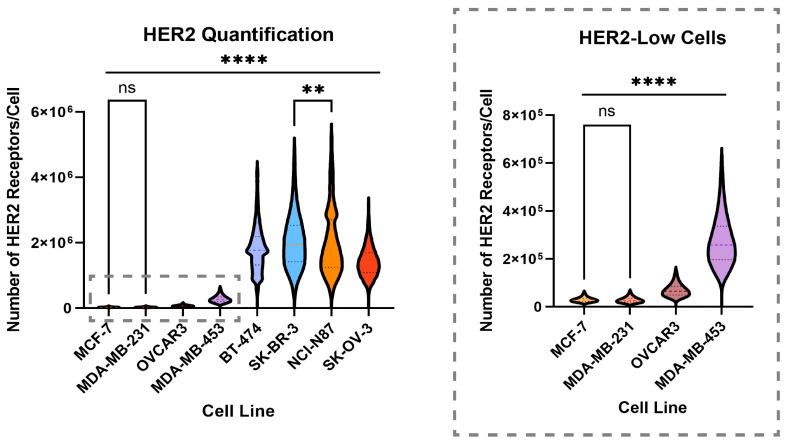
HER2 copy numbers on cell surfaces determined by flow cytometry. Violin plots demonstrate the distribution of HER2 expression among cell lines; this distribution is especially prevalent in the HER2-positive cell lines (BT-474, SK-BR-3, NCI-N87, and SK-OV-3). For each cell line, N = 9. Cells were fixed prior to staining with T-AF647 in excess. The passage numbers were 6–12. All data points from flow cytometry were analyzed, and outliers were identified with a Q = 1 test. Violin plots show the median (dashed line) and 95% CI (dotted lines). ns = not significant, ** *p* < 0.01, **** *p* < 0.0001, Tukey.

**Figure 2 pharmaceutics-16-00752-f002:**
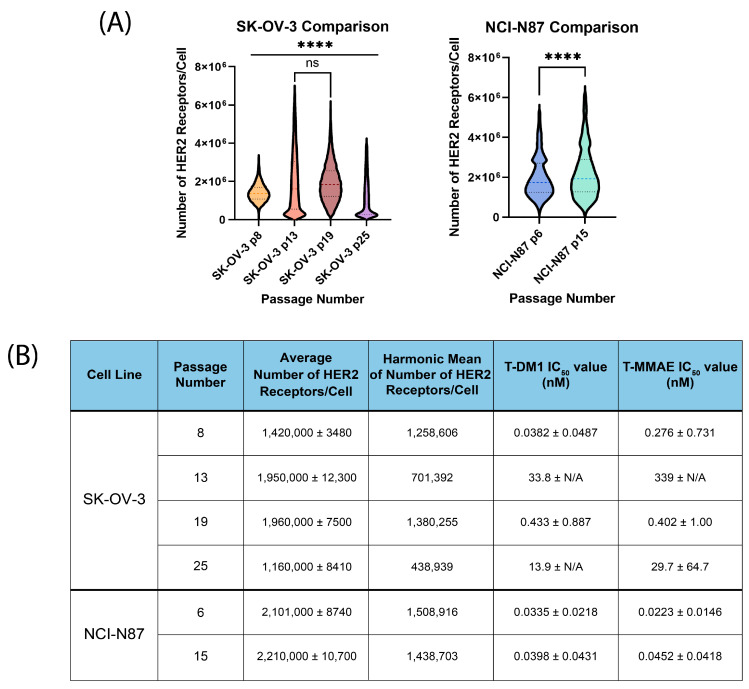
Passage number comparison for NCI-N87 and SK-OV-3 cell lines. (**A**) SK-OV-3 cell lines demonstrate an oscillating pattern of HER2 expression that corresponds with treatment response to HER2-targeted therapies. The NCI-N87 cell line does not follow this cyclic pattern, but HER2 expression increases with the passage number. All data points from the flow were analyzed, and outliers were identified with a Q = 1 test in GraphPad Prism. Violin plots show the median (dashed line) and 95% CI (dotted lines). ns = not significant, **** *p* < 0.0001. (**B**) Passage number comparison for NCI-N87 and SK-OV-3 cell lines. The SK-OV-3 cell line had variable HER2 expression among the passage numbers; the oscillating effect of both T-DM1 and T-MMAE matches the distribution pattern seen in A and the harmonic mean. NCI-N87 cell line had a slight increase in HER2 expression with increasing passage number. The large value and variability of T-MMAE IC_50_ values are due to the large distribution of receptors and the inability to reach 0% viability in assays. However, the treatment response to T-DM1 and T-MMAE and the harmonic mean between the two passage numbers were comparable. N ≥ 3 for all passages.

**Figure 3 pharmaceutics-16-00752-f003:**
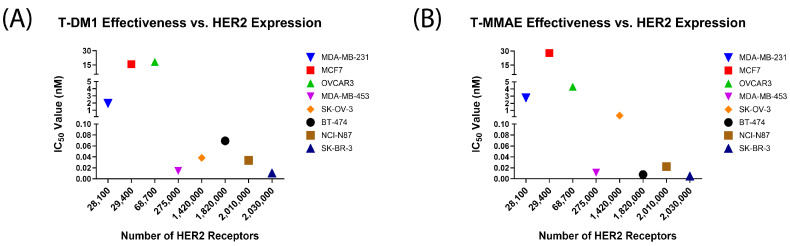
Correlation between the IC_50_ values of trastuzumab-based ADCs and the number of HER2 receptors. Each point represents the average of 9 replicates for the IC_50_ calculations (N = 9). (**A**) The correlation of trastuzumab–emtansine with a non-cleavable SMCC linker (T-DM1) with the number of HER2 receptors per cell. Pearson correlation coefficient r = −0.639. Linear curve fit R^2^ = 0.408, exponential curve fit R^2^ = 0.265. (**B**) The correlation of trastuzumab–monomethyl auristatin E with a cleavable valine–citrulline linker (T-MMAE) with the number of HER2 receptors per cell. Pearson correlation coefficient r = −0.509. Linear curve fit R^2^ = 0.259, exponential curve fit R^2^ = 0.480. The linear curve goodness-of-fit and Pearson’s correlation coefficients do not show a strong linear relationship between the HER2 level and treatment efficacy in either case.

**Figure 4 pharmaceutics-16-00752-f004:**
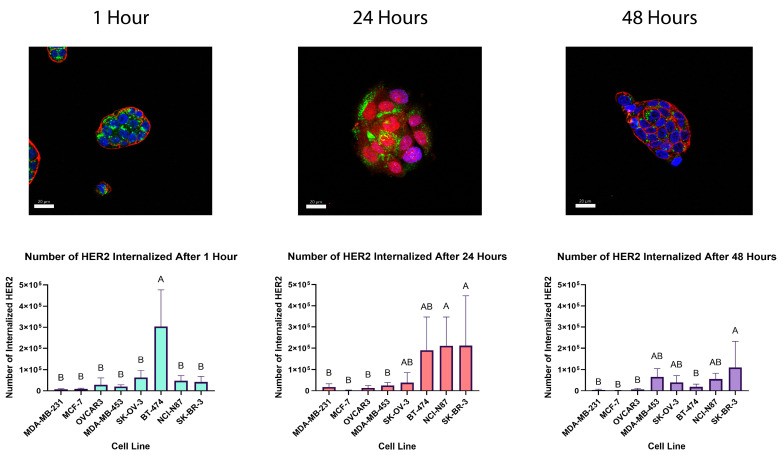
Representative confocal images of the HER2-positive NCI-N87 cell line stained with trastuzumab to label HER2 (red), anti-LAMP2 antibodies to label the lysosomes (green), and DAPI to label the nuclei (blue). The white bar in the bottom left corner of the images designates 20 µm. The colocalization of HER2 with the lysosomes was performed using the colocalization function in IMARIS version 10.0.1 software. The number of HER2 receptors internalized is based on the percentage of total HER2 determined by flow cytometry. Graphs show the calculated number of HER2 receptors colocalized with the lysosomes after the reported number of hours of treatment with trastuzumab. N = 3 or more for each cell line and time point. For the 1 h time point, every cell line labeled B is statistically different from BT-474 (A, *p* < 0.0001). For the 24 h time point, all cell lines labeled B are statistically different from cell lines labeled A (*p* < 0.02), while cell lines labeled AB are ns compared to all other cell lines. For the 48 h time point, cell lines labeled B are statistically different from SK-BR-3 (A, *p* < 0.02–0.007), while the cell lines labeled AB are not significantly different from SK-BR-3.

**Figure 5 pharmaceutics-16-00752-f005:**
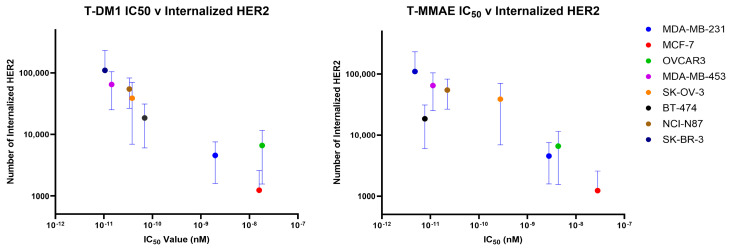
The correlation of the number of internalized HER2 receptors with IC_50_ values determined by GraphPad Prism version 9. A trend is observed showing that higher values of internalized HER2 result in lower IC_50_ values and more toxicity induced by the ADCs.

**Table 1 pharmaceutics-16-00752-t001:** Numerical data for the results shown in Figure 1 and Figure 2. N = 9 for each cell line. “N/A” for the error in T-MMAE IC_50_ values is due to the inability to reach 0% viability that results from the pile-growing tendencies of these cells.

Cell Line	Number of HER2 Receptors/Cell	T-DM1 IC_50_ Value (nM)	T-MMAE IC_50_ Value (nM)
MDA-MB-231	28,100 ± 89.1	1.97 ± 1.30	2.78 ± 2.64
MCF-7	29,400 ± 86.1	15.8 ± 18.92	27.9 ± N/A
OVCAR3	68,700 ± 209	18.3 ± 10.8	4.32 ± 3.37
MDA-MB-453	275,000 ± 794	0.0144 ± 0.0161	0.0113 ± 0.0186
SK-OV-3	1,420,000 ± 3480	0.0382 ± 0.0487	0.276 ± 0.731
BT-474	1,820,000 ± 5150	0.0692 ± 0.0760	0.00785 ± N/A
NCI-N87	2,010,000 ± 8740	0.0335 ± 0.0218	0.0223 ± 0.0146
SK-BR-3	2,030,000 ± 6360	0.0105 ± 0.00470	0.00481 ± 0.00256

## Data Availability

The original contributions presented in the study are included in the article/Appendix A, further inquiries can be directed to the corresponding author.

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
