# Peer review of "Surface-Available HER2 Levels Alone Are Not Indicative of Cell Response to HER2-Targeted Antibody–Drug Conjugate Therapies"

_pharmaceutics, 2024, doi:10.3390/pharmaceutics16060752_

Round 1

Reviewer 1 Report

Comments and Suggestions for Authors

            This excellently written, quite understandable and presented paper describes an in vitro  research on the efficacy of HER2-Targeted Antibody-Drug Conjugate Therapies, based on the analyses of eight  different cell lines, concluding that this efficacy “relies on both the number of HER2 proteins expressed and the level of HER2 internalization”. Consequently, “HER2 levels are not wholly adequate to determine response to ADC treatment” and “A more comprehensive patient assessment may be important in predicting response to HER2-targeting ADCs”.

Specific comments

1.      The authors seem to emphasize this study in relation to breast cancer but they have used cancer cell lines with other cancer origins such as ovarian or gastric carcinoma. At times, their observations and discussion do not make distinctions on this respect.   In this sense,  headings such as  Quantification of HER2 Expression on Representative Breast Cancer Cell Lines” andResponse of Breast Cancer Cell Lines to Trastuzumab Antibody-Drug Conjugates”  are confusing at first and could be refined.

2.      As an in vitro research with stablished cancer cell lines, these findings are to be taken into account but a major shortcoming and weakness of this study is how to determine the internalization rate of the ADCs in patients. The authors should suggest and/or comment something on this point. 

Reviewer 2 Report

Comments and Suggestions for Authors

The author demonstrated that HER2 levels expressed on the surface of cell lines are not wholly adequate to predict response to ADC treatment, while internalization ability also plays an essential role in providing effective therapy. The paper is generally well written and structured, especially with a good point as to highlight the importance of antibody trafficking and processing. Thus, I strongly recommend considering this article for publication in the Pharmaceutics journal. However, before acceptance, several issues should be addressed as follows:

1)    In Figure 2B, the average and/or harmonic mean number of HER2 receptors/cell from SK-OV-3 cells at passage 13 is much higher than the data at passage 25. But as referred to IC50 values, a reversed result was found with a big difference. Why?

2)    Is the reported T-MMAE IC50 value of 339 ± NA nM against SK-OV-3 cells accurate?

3)    In Figure 4, how did the author evaluate the internalized HER2 number into cells at different timepoints?

4)    Please keep the consistency in subtitle cases throughout the manuscript. 

Comments on the Quality of English Language

Moderate editing of English language required
